# Efficacy of processed amaranth-containing bread compared to maize bread on hemoglobin, anemia and iron deficiency anemia prevalence among two-to-five year-old anemic children in Southern Ethiopia: A cluster randomized controlled trial

**Alemselam Zebdewos Orsango** [1,2]*, **Eskindir Loha**[1,2], **Bernt Lindtjørn**[1,2], **Ingunn Marie S. Engebretsen**[2]

**1** School of Public Health, College of Medicine and Health Sciences, Hawassa University, Hawassa, Ethiopia, **2** Centre for International Health, University of Bergen, Bergen, Norway

* zalemselam@yahoo.com

## Abstract

### Background

Few studies have evaluated iron-rich plant-based foods, such as amaranth grain, to reduce anemia and iron deficiency anemia. Amaranth is rich in nutrients, but with high level of phytate. The objective of this trial was to evaluate the efficacy of home processed amaranth grain containing bread in the treatment of anemia, hemoglobin concentration and iron deficiency anemia among two-to-five year-old children in Southern Ethiopia.

### Method

Children with anemia (hemoglobin concentration <110.0g/L) (N = 100) were identified by random sampling and enrolled in a 1:1 cluster randomized controlled trial for six months in 2017. The amaranth group (N = 50), received 150g bread containing 70% amaranth and 30% chickpea, the amaranth grain was processed at home (soaking, germinating, and fermenting) to decrease the phytate level. The maize group (N = 50), received 150g bread, containing processed maize (roasted and fermented) to give a similar color and structure with amaranth bread. Hemoglobin, ferritin, and CRP were measured at baseline and at the end of intervention. Hemoglobin and ferritin values were adjusted for altitude and infection, respectively. Generalized estimating equation and generalized linear model were used to analyze the data.

### Result

In the last follow-up measure anemia prevalence was significantly lower in the amaranth group (32%) as compared with the maize group (56%) [adjusted risk ratios, aRR: 0.39 (95% CI: 0.16–0.77)]. Hemoglobin concentration estimate of beta coefficient was significantly

**Data Availability Statement:** All relevant data are uploaded to the OSF database and publicly accessible via the following URL: https://doi.org/10.17605/OSF.IO/EKUWZ.

**Funding:** Norwegian Program for Capacity Development in Higher Education and Research for Development/ South Ethiopia Network of Universities in Public Health (NORHED/SENUPH) provided funding for this study in the form of a grant awarded to BL (ETH-13/0025). The funders had no role in study design, data collection and analysis, decision to publish or preparation of the manuscript.

**Competing interests:** The authors have declared that no competing interests exist.

higher in the amaranth group compared with the maize group [aβ 8.9g/L (95%CI: 3.5–14.3)], p-value <0.01. The risk of iron deficiency anemia is significantly lower in the amaranth group [aRR: 0.44 (95%CI: 0.23–0.83)] in the intention to treat analysis but not significant in the complete case analysis. There was no significant difference between groups in iron deficiency [aRR: 0.81 (95%CI: 0.55–1.19)].

## Conclusion

Processed amaranth bread had favorable effects on hemoglobin concentration and has the potential to minimize anemia prevalence.

## Clinical trial registration

Trial registry number: PACTR201705002283263 https://pactr.samrc.ac.za/TrialDisplay.aspx?TrialID=2283

## Introduction

Chronic malnutrition is a leading health problem among children under five years of age in Ethiopia [1]. Anemia is a chronic type of malnutrition and a major contributor to childhood morbidity and mortality, limited physical and cognitive development, and decreased productivity later in life. In Ethiopia, 57% of children under five years old suffer from anemia [2]. The major cause of anemia is inadequate intake of micronutrients, mainly iron containing foods. These nutrients are largely found in animal-source food, which is generally unaffordable for families living in poverty. The staple food comprising the vast amount of energy in the diet of Ethiopians is dominated by starchy cereals, which are low in proteins, minerals, and vitamins [2].

Studies indicate that the amaranth plant has the potential to reduce food insecurity and resultant undernutrition [3,4]. Amaranth is a pseudo-cereal with high grain yield and capable of withstanding extreme climate and soil conditions. Amaranth also possesses good nutritional qualities with a high level of protein, minerals, and fat as compared to commonly utilized cereals, such as maize. Amaranth grain contains 16/100 grams (g) protein, 173 milligrams (mg)/100 g calcium, 35 mg/100 g iron, 3 mg/100 g zinc, and higher potassium, phosphorous, magnesium, manganese, vitamins A and E, and folic acid levels than cereal grains [5,6]. Consequently, it can constitute a cheaper alternative for poor households to address nutritional anemia.

Even though amaranth contains quality amino-acids, high iron and other micro-nutrients, it also has a high concentration of phytic acid that can reduce the bio-availability of nutrients, especially iron, protein, and zinc [6,7]. One study from Kenya indicated that amaranth has no significant effect on the anemia and iron status of children, and this could be due to a high iron to phytate molar ratio [6]. However, prior to this study, research had been performed on methods to decrease the phytate level of amaranth grain by utilizing home processing, such as soaking, germinating, and fermenting, using locally available materials to create acidic media. The research identified methods to decrease phytate level to an acceptable level of iron to phytate molar ratio of less than one [8,9] but its effectiveness has not yet been investigated *in vitro*. Thus, the present study aimed to evaluate the efficacy of processed amaranth grain compared

to the commonly consumed maize on hemoglobin concentration and on treatment of anemia prevalence among two-to-five year-old children in Southern Ethiopia.

## Methods

### Ethics approval and consent to participate

The Institutional Review Board of Hawassa University (IRB/098/08) provided ethical approval on September 06, 2016 and the Regional Ethical Committee West of Norway (No. 2016/2034) on December 15, 2016. Local administrative and health post authorities also granted official permissions for the study. Informed written consent was obtained from the mothers. The trial was registered (Pan African Clinical Trials Registry number: PACTR201705002283263). The authors confirm that all related trials for this food were registered.

The justification for delayed trial registration was a delay we faced when trying to register it at ClinicalTrial.gov. After getting the ethical approval, we immediately applied for the registration of the Clinical Trials on February 6, 2017, to ClinicalTrial.gov. Their response was unfortunately delayed, and we therefore dropped it on May 11, 2017. Then, we applied to Pan African Clinical Trial Registry (PACTR) on May 11, 2017 and shortly afterwards; we got the trial registration on May 19, 2017. During the time, we started the survey to maintain the study schedule. After the survey, we identified children with anaemia which required immediate interventions as per the ethical approval agreement.

### Protocol available

Protocol will be available after publication at: https://dx.doi.org/10.17504/protocols.io.bhzbj72n.

### Study area

The study was carried out in the Cheffe Cotte Jebessa kebele (lower administrative unit), in the suburban parts of Hawassa city, in southern Ethiopia. The altitude of the study area ranges from 1708 to 1920 meters above sea level. The annual average temperature is 12–27˚C with an average rainfall of 500 to 1,800 millimeter. The main crops cultivated in the area include maize, haricot beans, *enset*, and Irish potatoes. Amaranth grows as a wild crop in the study area, and its grains and leaves are used as food. Furthermore, the grain part is utilized for both medicinal purposes and alcoholic beverage production in the study area and other parts of the country.

### Study design

The study design was a community-based food intervention with a 1:1 cluster randomized controlled trial.

### Sample size

Prior to the trial, a census was conducted and 1689 children two-to-five year-old were identified in the study area. By using simple random sampling we selected 340 children and a cross-sectional study carried out to identify children with mild and moderate anemia. The sample size of 340 was estimated to identify anemic children, assuming a 33% mild and moderate anemia prevalence in South Ethiopia, precision of 0.5, and 95% confidence intervals. The sample size of the trial was calculated considering a mean difference of 0.7 in hemoglobin level [10], 1 standard deviation, power of 90%, alpha level of 0.05 with two sided test and an attrition rate of 12%. The total sample size of the trial was 100, with 50 children in each arm.

We conducted the survey from February 15, to March 30, 2017 and identified 100 children with anemia. The identified children were sparsely distributed in the study area and thus, we clustered them in to eight clusters by their geographic locations to ease the distribution of breads and for close follow-up. The eight clusters were randomly allocated into four clusters in the maize arm and four clusters in the amaranth arm, and each arm had 50 children. The random allocation was done by external body without knowing about the cluster. Then, the intervention (feeding of breads) was started for both arms on April 14, 2017 and continued for the next six months.

## Inclusion and exclusion criteria

Caregiver-child pairs were included in the study, in which the mother was the targeted respondent. The children included in the study were two-to-five year-olds with hemoglobin levels from 70 to <110 g/L, who lived in the area, and planned to live in the area for the next one year. Children were excluded if they were living with chronic illnesses, such as HIV/AIDS, tuberculosis, or cancer. Children who were taking iron supplements, received a blood transfusion in the last six months, had repeated malaria at least three times in the last three months, or were unwilling to participate in the study were also excluded.

## Outcome variables

The primary outcome was hemoglobin concentration and anemia prevalence, and the secondary outcome was iron deficiency anemia and iron deficiency.

## Intervention

In both groups, the intervention and the control, 150 g bread was provided to the children under the researcher team's supervision on a daily basis for the period of six months. The children in the experimental and control groups labelled 'amaranth group' and 'maize group', respectively. The amaranth group was supplied with bread containing 70 mass % amaranth grain and 30 mass % mashed chickpea. The maize group was supplied with the same size of bread containing 100% maize. All participants in the amaranth and maize groups were treated with albendazole 400 mg single dose, irrespective of their last dosage prior to the start of the feeding intervention and at the end of feeding intervention.

## Recipe preparation

Home level processing was applied to the amaranth grain to reduce phytate level. Amaranth grain was soaked in water by adding 5 ml of lemon juice per 100 ml of water for 24 hours and germinated for 72 hours. After sun drying, it was roasted and milled with a local electrical mill then fermented bread was prepared [9]. At the same time, maize grain was also roasted and fermented in order to make the recipe similar in color and structure with amaranth bread. The recipe was prepared based on the recommended dietary allowance (RDA); according to RDA, 150 g of bread (70% amaranth with 30% chickpea) had 22.3 mg of iron contained in the bread, which can fulfill 50% of RDA considering 15–20% iron absorption. The acceptability of this combination was tested in the community in the previous study and it was acceptable [9,10] "Table 1".

## Distribution and masking

The study participant, bread distributer, and data collector were uninformed about arm allocation and the content of the bread. Both grains, amaranth and maize, were processed (roasted

Table 1. The nutrient content of amaranth and maize bread.

| 100 g | Protein mg/100g | Fat mg/100g | Iron mg/100g | Calcium mg/100g | Zinc mg/100g | Energy Kcal |
|---|---|---|---|---|---|---|
| 100% maize | 7.58 | 4.44 | 4.31 | 24.82 | 2.47 | 386 |
| 70% amaranth and 30% chickpea | 16.39 | 7.08 | 15.16 | 244.71 | 2.59 | 404 |

mg, milligram; g, gram; Kcal, kilocalorie.

and fermented) to make the bread similar in color and structure. The bread distributers were different for the amaranth and maize arms, and there were eight caregivers, one for each cluster. Eight boxes, one for each cluster, were packed and labeled with the respective caregiver's name. Each caregiver feed the child at their home every day under her direct supervision. In case of refusals or absenteeism, unopened bread was returned and registered at coordinator level each day.

## Blood collection and laboratory methods

Three to five milliliter (ml) of venous blood was collected in test tubes with lithium heparin plasma to measure hemoglobin, serum ferritin, and CRP at the scheduled baseline and last follow-up visits. Hemoglobin concentration was determined immediately using a HemoCue analyser 301 (Angelholm, Sweden). All samples with hemoglobin <110 g/L were tested further for serum-ferritin and CRP. Anemia was classified based on altitude-adjusted hemoglobin concentration as normal ($\geq$110.0 g/L), mild (100.0–109.0 g/L), moderate (70.0–99.0 g/L), or severe (<70.0 g/L) [11,12]. Children with severe anemia (<70.0 g/L) identified in the survey were excluded from the trial, as they were referred for medical treatment (N = 1). Hemoglobin concentrations were corrected for altitudes, according to WHO standards [11,12].

Serum ferritin was analyzed using the Cobas 6000 e601 and CRP using the Cobas 6000 c501 module, both from Roche (Germany). Ferritin was adjusted for inflammation based on CRP measure. The inflammatory state of each individual was classified as "healthy" if CRP<5mg/L and "with inflammation" if CRP >5mg/L. The cut-off value for ferritin level was set at <12 μg/L for those children who had CRP<5mg/L (healthy) and <30 μg/L for those children who had CRP >5mg/L (inflammation). Iron deficiency defined as ferritin concentration <12 μg/L for healthy children and <30 μg/L for children with inflammation [13,14]. Iron deficiency anemia was defined as anemia with concurrent iron deficiency [14,15].

## Anthropometry measurements

Anthropometric weight and height were measured to assess nutritional status. Height was measured using a Seca 213 height board (Hamburg, Germany) with a sliding head piece while the child stood straight. Weight was taken using a calibrated Seca 874 electronic flat scale with the child barefoot and wearing light clothing. Anthropometry measurements were analysed according to the Emergency Nutrition Assessment for SMART software 2011 (Toronto, Canada), developed using WHO child growth standards [16]. Weight and height measurements were converted to height-for-age (HAZ), weight-for-age (WAZ), and weight-for-height (WHZ) z-scores, based on WHO reference standards. For WHZ, a z-score <-2 indicated wasting, $\geq$- 2 to <2 indicated normal, and >2 indicated overweight. For HAZ, a z-score <-2 indicated stunting, and >-2 indicated normal height. WAZ scores <-2 indicated underweight, and >-2 indicated normal. Moderate and severe undernutrition were defined as z-scores <-2 and <-3, respectively [17].

## Questionnaire

A 7 days structured food-frequency questionnaire (FFQ) and 24-hour dietary diversity questionnaire were used to collect information about children's diets. The foods were counted in nine food groups (items), in which the food-frequency items were created based on Food and Agriculture Organization (FAO) guidelines. The scores ranged from low (<3 food items), medium (4–5 food items), to high (≥6 food items) [18]. Dietary diversity and iron rich food (red meat, organ meat and fish) 7 days frequency was assessed every month to control confounding. The household food diversity scale was categorized into 12 food types, according to the Food and Nutrition Technical III project, and the answers were scored the same as for the food-frequency items [18]. We used a household food insecurity measure validated in Ethiopia by Gebreyesus [19]. The prevalence of household food insecurity was determined using the nine-component Household Food Insecurity Access Scale (HFIAS). Based on component scores (1 = rarely, 2 = sometimes, 3 = often), household food insecurity was classified as secure, mild, moderate, or severe [20].

## Statistical analysis

Data were double-entered and checked using EpiData v. 3.1 (Odense, Denmark), and transferred to IBM SPSS v. 20 (Chicago, IL, U.S.A.) and STATA 15 for analysis [21]. Descriptive statistics, including frequency counts and percentages, were used to summarize the data and chi-square test was used to compare the distribution of categorical variables in the amaranth and maize groups. Means, medians, confidence intervals (CI), and inter quartile ranges (IQR) were used to summarize continuous variables. Before multivariate analysis was performed, we checked for baseline differences of anemia, hemoglobin, and ferritin distributions between the study groups. Intra class correlation coefficients (ICC) was checked prior to running the multivariate analysis among the groups' 1:1 randomly allocated eight clusters: four clusters in the amaranth group and four clusters in the maize group. There was no significant intra class correlation for anemia, hemoglobin, and ferritin among group clusters (S1 Table).

Intention-to-treat analysis and complete case analysis were used to manage the missing value of main variables: hemoglobin, ferritin, and CRP (anemia, Iron deficiency anemia and iron deficiency). Intention-to-treat analysis (ITT) applied by replacing the missing values of the last follow-up measure from the baseline and a complete-case analysis was restricted to available information at the last follow-up measure was used to decrease miss-interpretation. A bivariate and multivariate generalized liner model for binomial family was used for anemia prevalence. Generalized estimating equations (GEE) procedure was used for analysis of repeated measurements (hemoglobin, ferritin and CRP) once at baseline and once at last follow up. Hemoglobin was not normally distributed and included as continuous variable. Adjusted ferritin with CRP was used to categorize iron deficiency anemia and iron deficiency so that it was included as categorical variable. Quasi likelihood under independency model criteria (QIC) with the lowest value was used to select correlation matrix. Child age, child sex, dietary diversity and stunting were included in the multivariate analysis. The estimate of beta coefficient with confidence interval was reported for continuous variables and relative risk ratio was presented for the categorical variables with confidence interval.

## Results

### Trial participants

Out of the 100 children in the trial, 82 children remained in the trial cohort at the end of the scheduled visits, 41children in the amaranth group and 41children in the maize group "Fig 1".

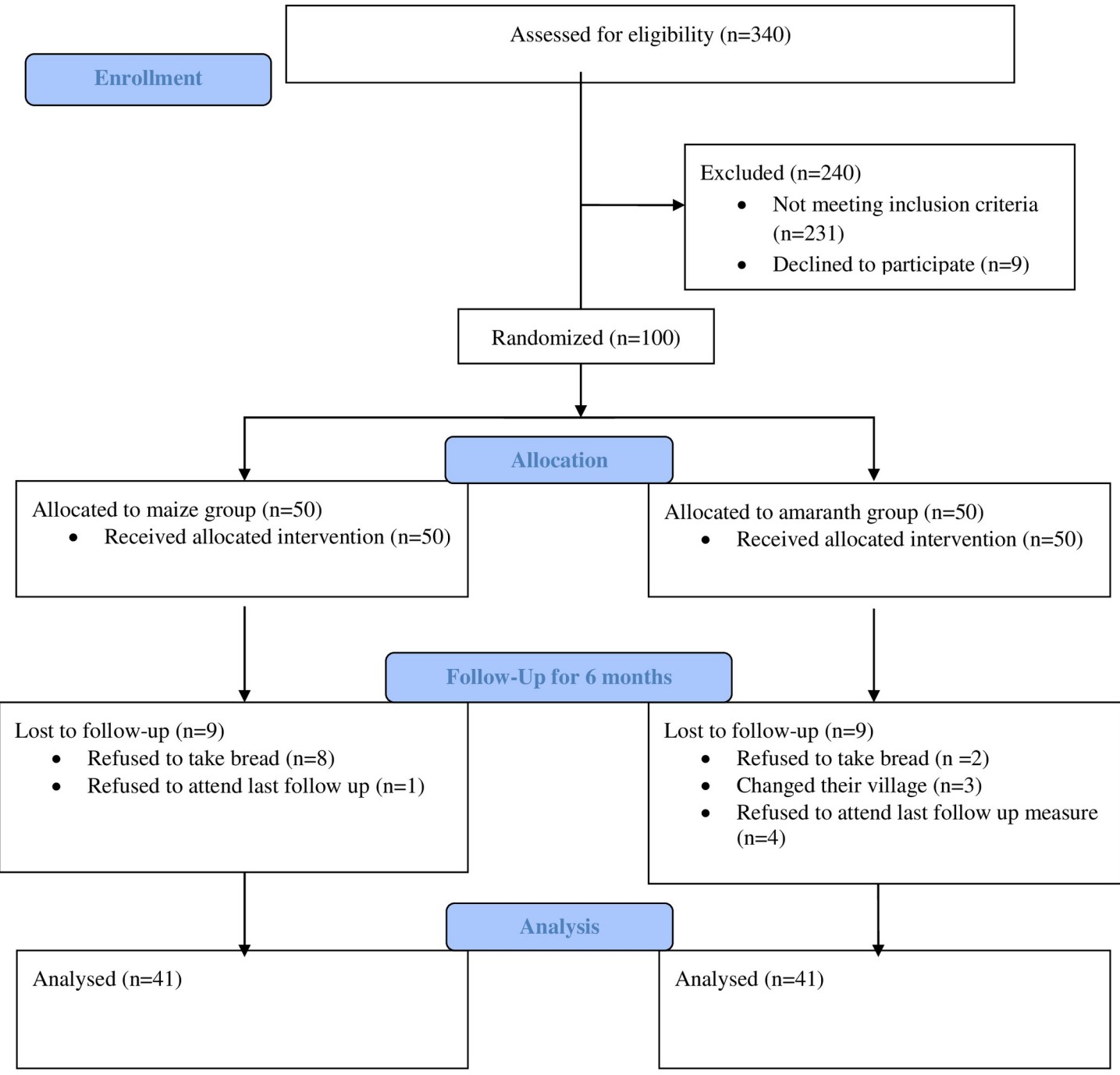

**Fig 1. Trial profile.**

There was no statistically significant difference were found in socio-demographic characteristics between discontinued (n = 18) and participated children at last follow-up measure (P>0.05) "S2 Table". The six months daily attendance of children with regard to bread consumption indicated that there was no significant mean difference between amaranth and maize groups [-5.32 (95% CI,-20.1; 9.5)] with P-value > 0.5 "Table 2".

**Table 2. Mean of bread consumption attendance between groups.**

| Six month children attendance | Mean (CI) | Mean difference (CI)e | P-value |
|---|---|---|---|
| Maize group | 146.9 (135.1–158.6) | -5.32 (-20.1;9.5) | 0.5 |
| Amaranth group | 152.2 (142.9–161.5) | | |

## Baseline characteristics of children

From the hundred participants, half were girls and the mean age was 37 months (95% CI: 34.8–39.1). Majority 73 (73%) of children scored the lowest dietary diversity and 4 (4%) were scored high dietary diversity. Child illness history indicated that 38 (38%) had a history of watery diarrhea in the past 15 days preceding the survey. Regarding the nutritional status of the children, 53 (53%) were stunted, and 28 (28%) were underweight. Stunting prevalence was significantly higher in the amaranth arm 31/50 (62%) than in the maize group 22/50 (44%) "Table 3".

## Baseline characteristics of mother and household

All caregivers were mothers, and their median age was 26 (IQR: 24–30) years. The median completed school years by mothers were 4 (IQR: 0–7) years and 63 (63%) of the mothers were unemployed. Majority 77 (77%) of the households were earned less than 1.9 USD per day, and 69 (69%) of the households were food insecure "Table 4".

## Magnitude of anemia and iron deficiency anemia at baseline

From the total children with anemia, 29/100 (29%) were moderately anemic and 71/100 (71%) were mildly anemic. Children with moderate anemia were 38% in amaranth group and 20% in maze group. Iron deficiency anemia constitutes 30% of the total anemia cases and distributed as 35% in the amaranth group and 24% in the maize group "Table 5".

## 24 hours dietary diversity score

The mean dietary diversity score (DDV) indicated that there was no significant difference between amaranth and maize arms for each month "Table 6".

## Six month follow-up of iron-rich food consumption

Seven days food frequency recall indicated that 59% of children's were not consumed iron-rich food at least one time for the last six consecutive months follow up. There is no significant difference between groups on iron-rich food consumption (p-value >0.5) "Table 7".

## The effect of amaranth containing bread on anemia prevalence

At last follow-up measure, the overall prevalence of anemia decreased to 44% as compared with 100% at baseline. Intention-to-treat analysis indicated that, children who received amaranth containing bread had 61% significant reduction in the risk of anemia compared to those who received maize bread [aRR: 0.39 (95%CI: 0.16–0.77)]. The prevalence of anemia was significantly lower (32%) in the amaranth group as compared with the maize group (56%) Similarly, the complete-case analysis result indicated a significant decrease of anemia prevalence in amaranth group (17%) as compared with (46%) in the maize group [aRR: 0.18 (95%CI: 0.05–0.57)], "Table 8".

**Table 3. Baseline characteristics of children, maize and amaranth groups.**

| Characteristics | Total N = 100 | Maize | Amaranth |
|---|---|---|---|
| **Child age month (mean (CI) (Continuous)** | 36.9 (34.8–39.1) | 36.6 (33.3–39.8) | 37.7 (34.9–40.5) |
| **Categorical data** | N (%)* | N = 50; N (%)* | N = 50; N (%)* |
| **Child sex** | | | |
| Boy | 50 (50) | 22 (44.0) | 28 (56.0) |
| Girl | 50 (50) | 28 (56.0) | 22 (44.0) |
| **Diarrhea in the past 15 days** | | | |
| No | 62 (62.0) | 32 (64.0) | 30 (60.0) |
| Yes | 38 (38.0) | 18 (36.0) | 20 (40.0) |
| **Cough in the past 15 days** | | | |
| No | 76 (76.0) | 37 (74.0) | 39 (81.3) |
| Yes | 22 (22.0) | 13 (26.0) | 9 (18.8) |
| Missing | 2 (2.0) | | |
| **Hospital admission since birth** | | | |
| No | 88 (88.0) | 43 (86.0) | 45 (90.0) |
| Yes | 12 (12.0) | 7 (14.0) | 5 (10.0) |
| **Child dietary diversity** | | | |
| Low | 73 (73.0) | 38 (77.6) | 35 (71.4) |
| Medium | 21 (21.0) | 10 (20.4) | 11 (22.4) |
| High | 4 (4.0) | 1 (1.0) | 3 (3.1) |
| Missing | 2 (2.0) | | |
| **Height for age (HAZ)** | | | |
| No stunting | 47 (47.0) | 29 (58.0) | 18 (36.0) |
| Stunting | 53 (53.0) | 22 (42.0) | 31 (64.0) |
| **Weight for height (WHZ)** | | | |
| Normal | 89 (89.0) | 44 (95.7) | 45 (88.0) |
| Wasting | 4 (4.0) | 1 (2.1) | 3 (6.0) |
| Overweight | 4 (4.0) | 1 (2.1) | 3 (6.0) |
| **Weight for age (WAZ)** | | | |
| Normal | 72 (72.0) | 38 (76.0) | 34 (68.0) |
| Underweight | 28 (28.0) | 12 (24.0) | 16 (32.0) |

## The effect of amaranth bread on hemoglobin concentration

Last follow-up measure indicated that hemoglobin concentration was increased in both amaranth and maize arms when compared with the baseline results. The estimate of beta coefficient indicated that children who received amaranth bread had significantly higher amount of hemoglobin concentration than maize bread, as the 95% CI does not contain zero [aβ 8.9 g/L (95%CI: 3.5–14.3)] "Table 9".

## The effect of amaranth bread on iron deficiency anemia

The overall prevalence of iron deficiency anemia decreased from 29% at baseline to 18% at last follow-up measure. Intention to treat analysis showed that iron deficiency anemia risk was significantly decreased in the amaranth group from 35% at baseline to 15% at the last follow-up [aRR: 0.44 (0.23–0.83)]. However, this difference was not significant in the complete case analysis "Table 10".

**Table 4. Baseline characteristics of mother and households.**

| Characteristics | Total N = 100 | Maize | Amaranth |
|---|---|---|---|
| Mother age (Continuous) median (IQR) | 26 (20–32) | 26 (19–31) | 26 (20–32) |
| Categorical data | N (%)* | N = 50; N %)* | N = 50; N (%)* |
| **Mother's occupation** | | | |
| Unemployed | 63 (63.0) | 32 (64.0) | 31 (62.0 ) |
| Day laborer | 23 (23.0) | 10 (20.0) | 13 (26.0) |
| Government | 14 (14.0) | 8 (16.0) | 6 (12.0) |
| **Father's occupation** | | | |
| Unemployed | 12 (12.0) | 7 (14.0) | 5 (10.0) |
| Daily laborer | 49 (49.0) | 20 (40.0) | 29 (58.4) |
| Government | 39 (39.0) | 23 (46.0) | 16 (32.0) |
| **Electricity access** | | | |
| No | 27 (27.0) | 13 (26.0) | 14 (28.0) |
| Yes | 73 (73.0) | 37 (74.0) | 36 (72.0) |
| **Household food security** | | | |
| Food secure | 31 (31.0) | 15 (30.0) | 16 (32.0) |
| Mild food-insecurity | 14 (14.0) | 6 (12.0) | 8 (16.0) |
| Moderate food-insecurity | 18 (18.0) | 10 (20.0) | 8 (16.0) |
| Severe food-insecurity | 37 (37.0) | 19 (38.0) | 18 (36.0) |
| **Household income per month** | | | |
| <1500 (1.9 dollars per day) | 77 (77.0) | 35 (71.4) | 42 (85.7) |
| >1500 (1.9 dollars per day) | 21 (21.0) | 14 (28.6) | 7 (14.3) |
| Missing | 2 (2.0) | | |

### The effect of amaranth bread on iron deficiency

In the amaranth group iron deficiency was decreased, from 34% at baseline measure to 26% at last follow-up measure. But the risk of iron deficiency had no significant difference between groups [aRR: 0.81(0.55–1.19)] "Table 11".

## Discussion

Research on the nutritional value of amaranth and methods to decrease phytate level has been performed [8,9]. However, few investigations exist focusing on the effect of processed amaranth *in vitro*. This study is the first in Ethiopia to assess the effect of homemade processed amaranth grain on the hemoglobin concentration, anemia and iron deficiency anemia status of children two-to-five years-old.

The findings indicated that hemoglobin concentration increased and the prevalence of anemia decreased significantly after intervention. The mean hemoglobin concentration change

**Table 5. Magnitude of anemia and iron deficiency anemia distribution between groups.**

| | Maize N (%) | Amaranth N (%) | Total N (%) |
|---|---|---|---|
| **Iron deficiency anemia N = 91** | | | |
| Yes | 11 (24.4) | 16 (34.8) | 27.0 (29.7) |
| No | 34 (75.6) | 30 (65.2) | 64.0(70.3) |
| **Anemia Total N = 100** | | | |
| Mild | 40 (80.0) | 31 (62.0) | 71.0 (71.0) |
| Moderate | 10 (20.0)) | 19 (38.0) | 29.0(29.0) |

**Table 6. The mean dietary diversity score of 24 hours recall.**

| Month | Group | Dietary diversity mean score (CI) | Mean Difference (CI) | P-value |
|---|---|---|---|---|
| 1 | Maize | 2.5 (2.6–2.2) | -0.24 (-.82;.33) | 0.4 |
| | Amaranth | 2.8 (2.3–3.3) | -0.24 (-.82;.33) | 0.4 |
| 2 | Maize | 2.5 (2.1–2.9) | -0.39 (-1.15;.36) | 0.3 |
| | Amaranth | 3.1 (2.5–3.7) | -0.39 (-1.15;.36) | 0.3 |
| 3 | Maize | 2.8 (2.4–3.2) | -0.49 (-1.19;.21) | 0.2 |
| | Amaranth | 3.2 (2.6–3.7) | -0.49 (-1.19;0.21) | 0.2 |
| 4 | Maize | 2.9 (2.4–3.3) | -0.04 (-0.66;0.58) | 0.8 |
| | Amaranth | 2.8 (2.3–3.2) | -0.04 (-0.66–0.58) | 0.8 |
| 5 | Maize | 2.9 (2.4–3.4) | -0.31 (-1.01–0.38) | 0.4 |
| | Amaranth | 3.1 (2.6–3.6) | -0.31 (-1.01–0.38) | 0.4 |
| 6 | Maize | 3.4 (2.9–3.9) | -0.26 (-0.94–0.41) | 0.4 |
| | Amaranth | 3.6 (3.2–4.1) | -0.27 (-0.94–0.41) | 0.4 |

was significantly higher in the amaranth arm as compared with maize arm. Also the risk of anemia was significantly lower in the amaranth arm as compared with maize arm. There is currently no extant literature in the study area examining the effect of processed amaranth grain on anemia status to which our results can be compared. However, one study conducted on amaranth leaf in Ghana showed similar result to this study. Amaranth leaf dried powder had a significant effect on anemia prevalence. Specifically, anemia prevalence decreased by 28% and hemoglobin concentrations were higher by 8.9 g/L after using amaranth leaf powder [10]. The presence of micronutrients other than iron in the amaranth grain, such as folic acid, copper, and vitamin A may have contributed to the increased hemoglobin [22–24], but changes in those levels were not quantified either in that study or the present one.

In contrast, a study conducted in Kenya demonstrated that amaranth did not decrease anemia prevalence significantly [6]. But the study used raw/unprocessed amaranth grain, which is high in phytate, and suggested that reducing the phytate level of amaranth grain may contribute to increasing the hemoglobin level. It is widely recognized that phytate can impede the absorption of iron and other micro- and macro-nutrients from the gut. This problem can be resolved, however, by applying homemade processing, such as soaking, germinating, and fermenting [8,9]. Therefore, we believe that the result obtained in this study was related with homemade processing applied to amaranth grain that decreased the phytate level and enhanced the absorption of iron and other micro- and macro-nutrients.

Iron deficiency anemia is significantly decreased in the amaranth group in the last follow up measure in the intention to treat analysis. But the change was not significant in the complete case analysis. This difference may be due to the presence of higher inflammation at baseline, confirmed by high CRP level in both arms when compared with last follow up. This is supported by other studies reporting that CRP and ferritin have a positive correlation [25]. This means some of the ferritin level observed as normal at baseline measure may related with

**Table 7. Seven days iron-rich food frequency recall of six consecutive months follow-up.**

| Cumulative Frequency | Maize | Amaranth | Total | P-value |
|---|---|---|---|---|
| 0 | 27 (65.9) | 22 (53.7) | 49 (59.8) | 0.7 |
| 1 | 8 (19.5) | 9 (22.0) | 17 (20.7) | |
| 2 | 4 (9.8) | 6 (14.6) | 10 (12.2) | |
| 3 | 2 (4.9) | 4 (9.8) | 6 (7.3) | |

**Table 8. The effect of amaranth bread on anemia prevalence.**

|  | No anemia | Anemia | CRR (95% CI) | ARR (95%CI) |
|---|---|---|---|---|
| **Intention-to-treat analysis** |  |  |  |  |
| Maize | 22/50 (44.0) | 28/50 (56.0) | 1 | 1 |
| Amaranth | 34/50 (68.0) | 16/50 (32.0) | 0.57 (0.35–0.92) | 0.39 (0.16–0.77) |
| **Complete-case analysis** |  |  |  |  |
| Maize | 22/41 (53.7) | 19/41 (46.3) | 1 | 1 |
| Amaranth | 34/41 (82.9) | 7/41 (17.1) | 0.37 (0.17–0.78) | 0.18 (0.05–0.57) |

CRR, crude relative risk; ARR, adjusted relative risk; CI, confidence interval; Adjusted for child age, child sex, and child nutritional status, height for age, dietary diversity. For the intention-to-treat analysis, all missing outcomes were treated as a failure (i.e., considered as anemic without knowing their status). For the complete-case analysis, only individuals with outcome data available were included.

the presence of inflammation. We adjusted ferritin with CRP level but this may not control the false reading of ferritin by hundred percent [26]. This condition may prevent discernment of the precise effect of amaranth on ferritin level or iron deficiency anemia. In the future, larger sample size and inclusion of additional iron indicator tests may provide more information on the iron status of children [26].

The last follow up measure indicated that anemia prevalence was noticeably decreased in the maize group, as well. This may be attributable to a reduction of helminth burden due to albendazole given to both arms. This result also supported by research conducted on the effect of deworming on anemia prevalence [27,28]. Furthermore, the daily supervision of a bread provider at the household level may have created awareness of sanitation and child-feeding practices, which in turn may have had a positive impact by decreasing the incidence of inflammation and improved dietary diversity. The fermented maize bread itself may also have increased the intake of iron and other micro and macro nutrients.

The acceptability test was not done for this particular research. However, prior to this study a similar combination was tested for acceptability in the research area by the same author and the result indicated the prepared food was acceptable by the community [9]. Non-acceptability of the food was also not reported during supervision and follow-up time.

Regarding the iron intake, there were no other iron supplementation programs in the research area. Further, we did a follow up every month for dietary diversity and iron-rich and bioavailable food consumption such as animal product (red meat, organ meat and fish). The six month follow up indicated that there was low consumption of iron-rich food and there was no significant difference between groups both for dietary diversity score and iron-rich food consumption.

**Table 9. The effect of amaranth bread on hemoglobin concentration.**

|  | Baseline mean Hemoglobin g/L | Last follow up mean Hemoglobin g/L | Estimated of beta coefficient (CI) | P-value |
|---|---|---|---|---|
| **Complete case analysis** |  |  |  |  |
| Maize | 103.5 (100.9–106.0) | 110.1 (106.9–113.2) | 1 |  |
| Amaranth | 101.3 (98.7–103.9) | 116.2 (112.1–120.3) | 8.9 (3.5–14.3) | 0.003 |
| **Intention to treat analysis** |  |  |  |  |
| Maize | 103.7 (101.5–105.8) | 109.1 (106.4–111.8) | 1 |  |
| Amaranth | 101.0 (98.7–103.4) | 113.3 (109.4–117.1) | 7.0 (2.1–12.0) | 0.006 |

Hemoglobin was adjusted for time, sex, age, and nutritional status (height for age).

CI: confidence interval; estimated beta coefficient, CI, and P-value analyzed using the GEE linear model.

**Table 10. The effect of amaranth bread on iron deficiency anemia.**

| | Baseline | | Last measurement | | Adjusted relative risk ratio (95% CI) |
|---|---|---|---|---|---|
| | Iron deficiency anemia | | Iron deficiency anemia | | |
| Complete case analysis N = 76 | No | Yes | No | Yes | |
| Maize | 30 (78.9) | 8 (21.1) | 30 (78.9) | 8 (21.1) | 1 |
| Amaranth | 25 (65.8) | 13 (34.2) | 31 (81.6) | 7 (18.4) | 0.54 (0.28–1.03) |
| ITT analysis N = 91 | | | | | |
| Maize | 34 (75.6) | 11 (24.4) | 35 (77.8) | 10 (22.2) | 1 |
| Amaranth | 30 (65.2) | 16 (34.8) | 39 (84.8) | 7 (15.5) | 0.44 (0.23–0.83)* |

CI: confidence interval; relative risk ratio, CI, and P-value analyzed using the GEE binomial model. Iron deficiency anemia was adjusted for time, sex, age, dietary diversity and nutritional status (height for age).

## Strengths and limitations of the study

This study used a simple random method to identify the study participants and cluster randomization to allocate anemic children into the two arms, so that the results obtained in this research can be reproducible if performed in a similar manner. The bread maker, bread distributor, caregivers, and data collectors were unaware of the maize or amaranth bread allocations. The caregivers fed their children daily under direct supervision. Flour blending was performed carefully considering the ratio of amaranth, and the ratio and size of the bread were checked regularly by a trained supervisor.

This study does not assessed other nutrient-related anemia, such as folic acid deficiency, vitamin A deficiency, and copper deficiency as well as does not considered genetic related hemoglobin disorder even though it is not common in our setup. Further we didn't measured AGP (a-1-acid glycoprotein) concentration to adjust ferritin. In addition, we did not conducted malaria test and stool examination The sample size for the survey was established when individual randomization was planned; however, during the survey period, we understood that children with anemia sparsely distributed and we clustered them by geographical location to effectively manage home based feeding and supervision, and thus clustering was not taken into account during sample size calculations however we managed during analysis using intra class correlation coefficient.

## Conclusion

The consumption of processed amaranth-containing bread decreased anemia prevalence, increased mean hemoglobin concentration, and minimized iron deficiency anemia prevalence

**Table 11. Iron deficiency distribution between groups at baseline and at last follow-up measure.**

| Intervention group | Iron deficiency Baseline | | Iron deficiency last follow up | | Relative risk (95%CI) |
|---|---|---|---|---|---|
| | Yes | No | Yes | No | |
| Complete case analysis N = 76 | | | | | |
| Maize | 8 (21.1)) | 30 (78.9) | 9 (23.7) | 29 (76.3) | 1 |
| Amaranth | 13 (34.2) | 25 (65.8) | 10 (26.3) | 28 (73.7) | 1.12 (0.81–1.55) |
| Intention to treat analysis N = 91 | | | | | |
| Maize | 11 (24.4) | 34 (75.6) | 12 (26.7) | 33 (73.3) | 1 |
| Amaranth | 16 (34.8) | 30 (65.2) | 13 (28.3) | 33 (71.7) | 0.81 (0.55–1.19) |

CI: confidence interval; relative risk ration and CI; analyzed using the GEE binomial model. Iron deficiency were adjusted for time, sex, age, dietary diversity and nutritional status (height for age).

of the participated children. Therefore, processed amaranth grain bread has the potential to minimize the prevalence of anemia.

## Supporting information

**S1 Table. Main outcome variables result of intra class correlation coefficient.**
(DOCX)

**S2 Table. Comparing the sociodemographic distribution between complete case and missed case.**
(DOCX)

**S1 Appendix. Questionnaire used to collect data.**
(DOCX)

**S2 Appendix. Trial protocol.**
(DOCX)

**S3 Appendix. CONSORT checklist.**
(DOCX)

## Acknowledgments

The authors are grateful to Mrs. Abebayehu Zebdewos and Mr. Teketel for blood sample collection in the field. Mr. Damte Data Daba and Mrs.Kassaye Zebdewos Orsango were coordinators of field activities. All data collectors performed data collection in the field. We also want to thank the child caregivers, drivers, bread makers, the Cheffe Cotte Jebessa Health Center, all health post workers, Cheffe Cotte Jebessa community leaders, and all study participants.

## Author Contributions

**Conceptualization:** Alemselam Zebdewos Orsango, Ingunn Marie S. Engebretsen.

**Data curation:** Alemselam Zebdewos Orsango, Bernt Lindtjørn, Ingunn Marie S. Engebretsen.

**Formal analysis:** Alemselam Zebdewos Orsango, Ingunn Marie S. Engebretsen.

**Funding acquisition:** Eskindir Loha, Bernt Lindtjørn.

**Investigation:** Alemselam Zebdewos Orsango, Ingunn Marie S. Engebretsen.

**Methodology:** Alemselam Zebdewos Orsango, Eskindir Loha, Bernt Lindtjørn, Ingunn Marie S. Engebretsen.

**Project administration:** Alemselam Zebdewos Orsango, Eskindir Loha, Bernt Lindtjørn.

**Resources:** Eskindir Loha, Bernt Lindtjørn.

**Software:** Alemselam Zebdewos Orsango, Eskindir Loha, Bernt Lindtjørn, Ingunn Marie S. Engebretsen.

**Supervision:** Alemselam Zebdewos Orsango, Eskindir Loha, Bernt Lindtjørn, Ingunn Marie S. Engebretsen.

**Validation:** Alemselam Zebdewos Orsango, Bernt Lindtjørn, Ingunn Marie S. Engebretsen.

**Visualization:** Alemselam Zebdewos Orsango, Eskindir Loha, Bernt Lindtjørn, Ingunn Marie S. Engebretsen.

**Writing – original draft:** Alemselam Zebdewos Orsango, Ingunn Marie S. Engebretsen.

**Writing – review & editing:** Alemselam Zebdewos Orsango, Eskindir Loha, Bernt Lindtjørn, Ingunn Marie S. Engebretsen.

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
