## [Decision Letter · Decision Letter 0]

14 May 2020

PONE-D-19-33677

To: Efficacy of processed amaranth bread on anemia prevalence, and change in C-reactive protein adjusted ferritin and hemoglobin levels among two-to-five year-old anemic children in Southern Ethiopia: A cluster randomized controlled trial

PLOS ONE

Dear Mrs Orsango,

Thank you for submitting your manuscript to PLOS ONE. After careful consideration, we feel that it has merit but does not fully meet PLOS ONE’s publication criteria as it currently stands. Therefore, we invite you to submit a revised version of the manuscript that addresses the points raised during the review process.

The manuscript has been evaluated by three reviewers, their comments are available below.

The reviewers have raised a number of major concerns, regarding the reporting of your manuscript, statistical analysis and terminology used. For example, they have requested a possible adjustment for additional confounders and further clarification of the sample size calculation.

Could you please carefully revise the manuscript to address all comments raised.

We would appreciate receiving your revised manuscript by Jun 27 2020 11:59PM. To enhance the reproducibility of your results, we recommend that if applicable you deposit your laboratory protocols in protocols.io, where a protocol can be assigned its own identifier (DOI) such that it can be cited independently in the future. For instructions see: http://journals.plos.org/plosone/s/submission-guidelines#loc-laboratory-protocols

We look forward to receiving your revised manuscript.

Kind regards,

Sara Fuentes Perez, PhD

Staff Editor

PLOS ONE

Journal Requirements:

2. Please address the following:

- Please refer to any post-hoc corrections to correct for multiple comparisons during your statistical analyses. If these were not performed please justify the reasons. Please refer to our statistical reporting guidelines for assistance (https://journals.plos.org/plosone/s/submission-guidelines.#loc-statistical-reporting).

- Please include additional information regarding the survey or questionnaire used in the study and ensure that you have provided sufficient details that others could replicate the analyses. For instance, if you developed a questionnaire as part of this study and it is not under a copyright more restrictive than CC-BY, please include a copy, in both the original language and English, as Supporting Information.

3. Thank you for submitting your clinical trial to PLOS ONE and for providing the name of the registry and the registration number. The information in the registry entry suggests that your trial was registered after patient recruitment began. PLOS ONE strongly encourages authors to register all trials before recruiting the first participant in a study.

1) your reasons for your delay in registering this study (after enrolment of participants started);

2) confirmation that all related trials are registered by stating: “The authors confirm that all ongoing and related trials for this drug/intervention are registered”.

Please also ensure you report the date at which the ethics committee approved the study as well as the complete date range for patient recruitment and follow-up in the Methods section of your manuscript.

5. Please amend either the title on the online submission form (via Edit Submission) or the title in the manuscript so that they are identical.

6. We note you have included a table to which you do not refer in the text of your manuscript. Please ensure that you refer to Table 6 in your text; if accepted, production will need this reference to link the reader to the Table.

Additional Editor Comments (if provided):

Reviewers' comments:

Reviewer's Responses to Questions

**Comments to the Author**

1. Is the manuscript technically sound, and do the data support the conclusions?

Reviewer #1: Partly

Reviewer #2: Partly

Reviewer #3: Partly

2. Has the statistical analysis been performed appropriately and rigorously? 

Reviewer #1: Yes

Reviewer #2: Yes

Reviewer #3: No

3. Have the authors made all data underlying the findings in their manuscript fully available?

Reviewer #1: Yes

Reviewer #2: Yes

Reviewer #3: Yes

4. Is the manuscript presented in an intelligible fashion and written in standard English?

Reviewer #1: No

Reviewer #2: Yes

Reviewer #3: No

5. Review Comments to the Author

Reviewer #1: A cluster randomized controlled trial was conducted to evaluate the efficacy of amaranth bread to treat anemia in young children in Southern Ethiopia and estimate changes in C-reactive protein, hemoglobin and ferritin. Anemia prevalence was lower post-intervention in the amaranth treated arm than the control arm.

Minor revisions:

1- Abstract: Provide the proportions instead of the fractions: 16/50 and 28/50.

2- Line 103: Indicate the statistical testing method which attains 90% power. State the alpha level and indicate if it was one- or two-sided.

3- Indicate the date range subjects were enrolled in the study.

4- Line 197: The chi-square test is used to compare rather than show a distribution of data.

5- Line 199: Replace the term “present” with “summarize.”

6- Line 207: Awkward wording: “was done.”

7- Line 208: Be more descriptive in this phrase, “end-line were set as anemic cases.” Consider changing end-line to last follow-up or the specific time point.

8- Line 213: Consider removing the sentence, “The repeated observation within one subject is not independent of each other.” Since this is implied by the previous sentence.

9- Line 217: Clarify if two measures were taken at baseline and another two were collected at end-line.

10- The sentence at line 218, “Consequently, a linear generalized …” seems to be redundant.

11- Line 221: Provide the rationale for selecting a matrix structure of independent.

12- Line 222: The “estimate” of the beta coefficient was reported.

13- Abstract: Clarify that the estimate of beta is reported in the abstract.

13- Line 229: The sentence beginning, “The number of boys..” is wordy making it difficult to understand.

14- Paragraph beginning at line 239: All these results can be shown in a table, and the paragraph can be made more concise.

15- All Tables: To improve clarity, consider format changes using double line spaces between the characteristics and single line spaces between the choices. Currently, it is difficult to distinguish where the results for one characteristic ends and the next one begins.

16- Paragraph beginning at line 258: Provide p-values or 95% confidence intervals to support the claim that the differences were significant.

17- Line 272: Indicate that the children who received amaranth bread had significantly higher amounts of hemoglobin which is indicated by a CI that does not contain zero.

18- Table 5: Replace “Beta coefficient” with “estimate of beta coefficient.”

Reviewer #2: This study sought to evaluate the efficacy of a staple grain-based food product, rich in iron, on "anemia" in young Ethiopian children.

General comment: "Anemia" simply refers to reduced HGB levels -- throughout the paper, if you're referring to iron deficiency anemia (IDA), please be clear about that. This is confusing because your paper is based on the delivery of an IRON-RICH food, which presumably would improve IRON STORES, thus reduce IDA. This food, however, would not reduce anemia due to other causes.

Line 155-169: In your Methods, you define Anemia and ID separately. But what ab classification of ID?

Who was anemic, without ID?

Who was IDA?

Who was ID, without anemia?

How this was analyzed / grouped is not reported in the Statistical analysis section.

Results

Line 257: The prevalence of anemia is unclear given the Qs above, and this needs to be addressed throughout the remainder of the paper. The classification of ID, IDA, etc need to be made.

Line 270: This is not showing an effect/impact of the Food on prevalence of ID with/w/out anemia -- these analysis need to be completed. In Table 5, you merely present effects on biochemical indicators. Why were these analyses not adjusted for inflammation?

Table 6 presents adjusted Fer only and presented in a different way. But why are the Results in Table 6 not shown adjusted?

Discussion: there is nothing that states acceptability of the foods consumed in this particular trial. What was compliance? FFQ Data -- what was their intake of similar foods? What was usual intake of Fe?

Reviewer #3: This study compares haemoglobin and ferritin response (C reactive protein adjusted) in iron rich Amaranth compared to Maize in a bread format in anaemic (<110 g/L) children 2-5 y. Results indicate a positive benefit of the Amaranth on haemoglobin, anaemia, but not ferritin. This is an important study as Amaranth has a high yield and grows well in challenging conditions. It is rich in micronutrient for a plant-based food but is high in phytates which limit micronutrient bioavailability; this study show despite that Hb is improved. However, I have some major concerns.

Comments

Abstract

I do think in the abstract in the objective you need to capture what you mean by processing. Amaranth grain that has been processed to reduce the effect of phytate. It is indicated later in the abstract how it was processed but it is not clear that the reason for this processing is to reduce phytates.

The Betas are confusing and maybe would be better presented as an adjusted mean difference with a 95% CI. A P value should be given. I assume the B is the difference between the difference for each treatment from baseline and end line

Introduction

Clear and well written. What is the primary outcome?

Methods

The sample size seems based on the continuous variable haemoglobin not on anaemia seems like it is the primary outcome in the introduction. Has the effect of clustering been taken into consideration in the sample size? The sample size seems small for this type of study.

Line 112 should g/dl preferably g/L not mg/dl

Please present haemoglobin in SI units g/L not g/dl.

Line 119 – what was the primary outcome

Maize contains phytate although less iron was the maize processed similarly to the Amaranth?

When was the blood collected at baseline relative to the start of the trial? Was there a screening (part of the larger survey) and then a baseline measure at the beginning of the trial.

Why did you use a categorical cut off for ferritin adjustment rather than Thurnam or BRINDA, this would have given you more sensitivity?

How was the bread dispensed? Did the children come daily to a village centre to receive the bread or was it packaged and sent home on a weekly basis? In the results section it looks

Line 205 Appendix 3 is the first of the appendices to be called out in the text where is 1 and 2.

Setting all individuals that had missing values at end line to anaemic is really just last value carried forward. Could you justify this? Clearly with regression to the mean and screening and including only those who were anaemic many of these would be non-anaemic at end line irrespective of treatment. I suppose one could presume this would be balanced across treatment. How were missing continuous outcome variables dealt with?

Line 203 – 205 Is this used to show that there was no effect of clustering ,yet clustering is included in the GEE. In the abstract it looks like the difference of difference was used to look at treatment effects. However, it looks like baseline values were included in the model. I get a bit lost in the stats there is a baseline measure and one follow-up measure?

Perhaps more clearly delineate how categorical and continuous dependent variable were dealt with.

Table 2 and 3. It is not appropriate to include P-Values when comparing treatment groups because if and difference exist they must be a chance finding. I believe you have adjusted for these in the models anyway which is appropriate. Is all the information required in Table 2 and 3 necessary? It would be appropriate to include the severity of anaemia by treatment group and low ferritin by treatment group.

The participants needed to anaemic at baseline (<110 g/L) for inclusion yet the mean Hb for maize is 111 g/L with the upper bound of the CI at 114 g/L. Also, the number of children is highly clustered around the cut-off 108-114 g/L. Is anaemia prevalence the correct outcome. This all leads me to believe that there was a screening and a baseline blood taken but this is not clear. Would Hb be more appropriate with a lot of people not anaemic at baseline how was dealt with in the analysis

I am really confused on the ferritin. You need to adjust the ferritin values using Thurnam or Brinda and then use some sort of linear regression. Table 6 doesn’t make sense to have three cut-offs, it should be the percentage with low iron based on the appropriate cut-off for ferritin.

Discussion

I don’t think Amaranth would contain any B12 Line 343. Also did you consider the presence of sickle cell or other genetic Hb disorders

6. PLOS authors have the option to publish the peer review history of their article (what does this mean?). If published, this will include your full peer review and any attached files.

Reviewer #1: No

Reviewer #2: Yes: Diane M. DellaValle

Reviewer #3: No

---

## [Author Response · Author response to Decision Letter 0]

26 Jul 2020

Dear Editor and reviewers 

We thank the editor and the three reviewers for their constructive comments which helped us to revise our manuscript. In the revised manuscript, we have carefully considered the editor’s and reviewers’ suggestions. We adjusted and added some paragraphs on statistical analysis, especially the issue with adjusting ferritin levels with standard methods. Furthermore, we addressed and responded to each of the points raised by the reviewers and editor. The reviewers’ and editor’s comments are below given in the bold format followed by our answer in normal font. 

Editor comment 

Answer

Thank you for the comment we ensure that we followed the PLOSONE style and all requirements in the manuscript.

2. Please refer to any post-hoc corrections to correct for multiple comparisons during your statistical analyses. If these were not performed please justify the reasons. Please refer to our statistical reporting guidelines for assistance (https://journals.plos.org/plosone/s/submission-guidelines.#loc-statistical-reporting).

Answer

Thank you for this suggestion. A post hoc correction was not done, as we had only two groups.

Answer

We included the survey and follow up questionnaire used in both original language and English as a supporting information. 

4. Thank you for submitting your clinical trial to PLOS ONE and for providing the name of the registry and the registration number. The information in the registry entry suggests that your trial was registered after patient recruitment began. PLOS ONE strongly encourages authors to register all trials before recruiting the first participant in a study.

1) your reasons for your delay in registering this study (after enrolment of participants started);

2) confirmation that all related trials are registered by stating: “The authors confirm that all ongoing and related trials for this drug/intervention are registered”.

Please also ensure you report the date at which the ethics committee approved the study as well as the complete date range for patient recruitment and follow-up in the Methods section of your manuscript.

Answer

Thank you for the comment. We justified the reason for delay of trial registration in the method part (L86-93)

The justification for delayed trial registration was a delay we faced when trying to register it at ClinicalTrial.gov. After getting the ethical approval, we immediately applied for the registration of the Clinical Trials on February 6, 2017, to ClinicalTrial.gov. Their response was unfortunately delayed, and we therefore dropped it on May 11, 2017. Then, we applied to Pan African Clinical Trial Registry (PACTR) on May 11, 2017 and shortly afterwards; we got the trial registration on May 19, 2017. During the time, we started to do the survey to maintain the study schedule. After the survey, we identified children with anaemia which required immediate interventions as per the ethical approval agreement.

The key dates and process detailed as follows 

On February 6, 2017 we submitted our first request to the ClinicalTrials.gov for trial registration 

On May11, 2017 we dropped our request from ClinicalTrials.gov due to substantial delay of the registration, and at the same date we applied to Pan African Clinical Trial Registry (PACTR) for the registration of trial

On May 19, 2017 we got our trial registration from Pan African Clinical Trial Registry (PACTR)

Answer

The data will be available if the paper is accepted at an open access web site, for example at osf.io

6. We note you have included a table to which you do not refer in the text of your manuscript. Please ensure that you refer to Table 6 in your text; if accepted, production will need this reference to link the reader to the Table.

Answer

Thank you for this comment and now we corrected it. 

Reviewer 1

Thank you very much for your constructive comment we learnt a lot from the comments and improved the manuscript. Based on comments we added the result, included key dates in the method part, and formatted the text and whole reports by avoiding redundancy and correcting awkward wording.

A cluster randomized controlled trial was conducted to evaluate the efficacy of amaranth bread to treat anemia in young children in Southern Ethiopia and estimate changes in C-reactive protein, hemoglobin and ferritin. Anemia prevalence was lower post-intervention in the amaranth treated arm than the control arm.

Minor revisions:

1. Abstract: Provide the proportions instead of the fractions: 16/50 and 28/50.

Answer

Thank you for your observation. We presented on the proportion, 32% on amaranth group and 56% on maize group (L 39)

2. Line 103: Indicate the statistical testing method which attains 90% power. State the alpha level and indicate if it was one- or two-sided.

Answer

Thank you for pointing this. Now we included alpha level of 0.05 with two sided test (L 114).

3. Indicate the date range subjects were enrolled in the study.

Answer

We accepted the comment and now we included the date range in the method part. We conducted the survey from Feb 15 to March 30, 2017. The intervention (feeding of breads) started on April 14, 2017 and continued for the next six months (L 116,122).

4. Line 197: The chi-square test is used to compare rather than show a distribution of data

Answer

Thank you we accepted the comment and corrected it as follow. Descriptive statistics, including frequency counts and, percentages were used to summarize the data and chi-square test was used to compare the distribution of characteristics in the amaranth and maize group (L 208-211).

5. Line 199: Replace the term “present” with “summarize

Answer

The term “Present” replaced by “summarize” (L 212).

6. Line 207: Awkward wording: “was done.”

Answer

Thank you for this comment. We rephrased it with the following wording: Intention-to-treat analysis and complete case analysis were used to manage the missing value of main variables: haemoglobin, ferritin, and CRP (L 218-220).

7. Line 208: Be more descriptive in this phrase, “end-line were set as anemic cases.” Consider changing end-line to last follow-up or the specific time point.

Answer

Thank you for the comment. We replaced “end line” by “last follow-up” throughout the document (L 222).

8. Line 213: Consider removing the sentence, “The repeated observation within one subject is not independent of each other.” Since this is implied by the previous sentence

Answer

We accept the comment, and we removed the sentence

9. Line 217: Clarify if two measures were taken at baseline and another two were collected at end-line. 

Answer

….procedure was used for analysis of repeated measurements (haemoglobin, ferritin and CRP) once at baseline and once at last follow up (L 224-225).

10. The sentence at line 218, “Consequently, a linear generalized …” seems to be redundant.

Answer

Thanks, we rephrased it with: Generalized estimating equations (GEE) procedure was used for analysis of repeated measurements (hemoglobin ferritin and CRP) once at baseline and once at last follow-up measure. Hemoglobin was not normally distributed and included as continuous variable. Adjusted ferritin with CRP was used to categorize iron deficiency anemia and iron deficiency, so that it was included as categorical variable (L 224-227).

11. Line 221: Provide the rationale for selecting a matrix structure of independent.

Answer

Thank you. Quasi likelihood under independency model criteria (QIC) with the lowest value was used to select correlation matrix (independent) (L 228-229).

12. Line 222: The “estimate” of the beta coefficient was reported.

Answer

Thank you for the comment. We replaced Beta coefficient with ‘’estimate of beta coefficient’’ (L 230).

13. Abstract: Clarify that the estimate of beta is reported in the abstract.

Answer

We have now described this and included it in the abstract section as: Hemoglobin concentration estimate of beta coefficient was significantly higher in the amaranth group compared with maize group [aβ 8.9 g/L (95%CI: 3.5-14.3)], p-value <0.01 (L 39-41). 

14. Line 229: The sentence beginning, “The number of boys.” is wordy making it difficult to understand.

Answer

Thank you for the comment. Considering the other reviewer’s comment we omitted the sentence which contains the phrase ‘‘number of boys ’’ and instead we described as follows: Out of the 100 children in the trial, 82 children remained in the trial cohort at the end of the scheduled visits, 41 children in the amaranth group and 41children in the maize group (Fig.1) (236-237).

15. Paragraph beginning at line 239: All these results can be shown in a table, and the paragraph can be made more concise. 

Answer

Thank you for the comment. We have now reduced the paragraph in the results (L 253-257).

16. All Tables: To improve clarity, consider format changes using double line spaces between the characteristics and single line spaces between the choices. Currently, it is difficult to distinguish where the results for one characteristic ends and the next one begins.

Answer

Thank you very much for pointing this. We did accordingly for all tables.

17. Paragraph beginning at line 258: Provide p-values or 95% confidence intervals to support the claim that the differences were significant

Answer

We accepted the comment and provided a 95% confidence intervals [aRR: 0.39 (95%CI: 0.16-0.77)] (L281) (L296).

18. Line 272: Indicate that the children who received amaranth bread had significantly higher amounts of hemoglobin which is indicated by a CI that does not contain zero.

Answer

The comment is accepted and corrected as follow: The estimate of beta coefficient indicate that the children who received amaranth bread had significantly higher amount of haemoglobin concentration than maize bread as the 95% CI does not contain zero 8.9 g/L (95%CI: 3.5-14.3) (L 293-296).

19. Table 5: Replace “Beta coefficient” with “estimate of beta coefficient

Answer

Thank you. We replaced ‘‘Beta coefficient’’ with ‘‘estimate of beta coefficient’’ (L 297)

Reviewer 2

This study sought to evaluate the efficacy of a staple grain-based food product, rich in iron, on "anemia" in young Ethiopian children.

Thank you very much for your constructive comments which helped us improves the manuscript. Based on your comment we clarified on the distribution of iron deficiency, iron deficiency anemia and iron deficiency. In addition, we describe the compliance of the participant and the measures we did during follow up on iron rich food frequency and dietary diversity to control for confounding.

1. General comment: "Anemia" simply refers to reduced HGB levels -- throughout the paper, if you're referring to iron deficiency anemia (IDA), please be clear about that. This is confusing because your paper is based on the delivery of an IRON-RICH food, which presumably would improve IRON STORES, thus reduce IDA. This food, however, would not reduce anemia due to other causes. 

Thank you very much for this comment. Yes, we assessed the effect of iron-rich food in treatment of anemic children. We used hemoglobin level to identify children with anemia, and ferritin level to identify Iron deficiency (ID) and Iron deficiency anemia (IDA). Therefore, with certain limitations, this research provided the change in hemoglobin concentration, anemia prevalence, iron deficiency and iron deficiency anemia by comparing the baseline measure with last measures. But we had a small sample size to conclude on the efficacy of the prepared recipe on iron deficiency anemia and iron deficiency. However, we think can make valid conclusions about hemoglobin change on the treated children. To clarity, we differentiated anemia, iron deficiency anemia, and iron deficiency in the study.

2. Line 155-169: In your Methods, you define Anemia and ID separately. But what ab classification of ID? Who was anemic, without ID? Who was ID, without anemia? How this was analyzed / grouped is not reported in the Statistical analysis section.

Answer

Thank you very much for pointing this. Now we described the classification of iron deficiency clearly in the method part. 

Ferritin was adjusted for inflammation based on CRP measure. The inflammatory state of each individual was classified as “healthy” if CRP<5 mg/L and “with inflammation” if CRP >5mg/L. The cut-off value for ferritin level was set at <12 µg/L for those children who had CRP<5mg/L (healthy) and <30 µg/L for those children who had CRP >5mg/L (inflammation). Iron deficiency defined as ferritin concentration <12 µg/L for healthy children and <30 µg/L for children with inflammation (1, 2). Iron deficiency anemia was defined as anemia with concurrent iron deficiency (2, 3) (L174-180).

3. Results

3.1. Line 257: The prevalence of anemia is unclear given the Qs above, and this needs to be addressed throughout the remainder of the paper. 

Thank you very much for the comment we clarified anaemia prevalence. At baseline, all study participants were children with anemia and we included only mild and moderate anemia cases. Now, this is described in the result part (Table 5) (L 264).

Please see on Table 5 Magnitude of anemia and iron deficiency anemia distribution between groups (L 264)

3.2. The classification of ID, IDA, etc need to be made.

Table 6 presents adjusted Fer only and presented in a different way. But why are the Results in Table 6 not shown adjusted? 

Answer

Thank you. Now we clarified ID and IDA. 

Ferritin was adjusted for inflammation based on CRP measure. The inflammatory state of each individual was classified as “healthy” if CRP<5 mg/L and “with inflammation” if CRP >5mg/L. The cut-off value for ferritin level was set at <12 µg/L for those children who had CRP<5mg/L (healthy) and <30 µg/L for those children who had CRP >5mg/L (inflammation). Iron deficiency defined as ferritin concentration <12 µg/L for healthy children and <30 µg/L for children with inflammation. Iron deficiency anemia was defined as anemia with concurrent iron deficiency. (L174-180) (Table 10 and 11) 

Please see on Table 10 The effect of amaranth bread on Iron deficiency anaemia (L 307) and 

Table 11 Iron deficiency distribution between groups at baseline and at last follow up measure (L 314)

4. Discussion: 

4.1. There is nothing that states acceptability of the foods consumed in this particular trial. What was compliance?

Thank you for your comment now we described in the result and discussion part 

Result

Study participant 

Out of the 100 children in the trial, 82 children remained in the trial cohort at the end of the scheduled visits, 41children in the amaranth group and 41children in the maize group “Fig.1”. There was no statistically significant difference were found in socio-demographic characteristics between discontinued (n=18) and participated children at last follow-up measure (P>0.05) “S2 Table”. The six months daily attendance of children with regard to bread consumption indicated that there was no significant mean difference between amaranth and maize groups [-5.32 (95% CI,-20.1; 9.5)] with P-value > 0.05 “Table 2”.(236-242).

Please see on Table 2 Six month children bread consumption attendance (L244)

Discussion

The acceptability test was not done for this particular research. However, prior to this study a similar combination was tested for acceptability in the research area by the same author and the result indicated the prepared food was acceptable by the community (4). In the current study non-acceptability of the food was not reported during supervision and follow-up time (L364-367). 

4.2. FFQ Data -- what was their intake of similar foods? What was usual intake of Fe?

Result 

Six month follow up of mean dietary diversity score (L266-268)

The mean dietary diversity score (DDV) indicated that there was no significant difference between amaranth and maize arms for each month “Table 6”.

Please see on Table 6 Dietary diversity score of 24 hours recall (L 270) 

Six month follow up of iron-rich food consumption (L 271-274)

Seven days food frequency recall indicated that 59% of children’s were not consumed iron-rich food at least one time for the last six consecutive months follow up. There is no significant difference between groups on iron-rich food consumption (p-value >0.5) “Table 7”.

Please see Table 7 Seven days iron rich food frequency recall of six consecutive follow-up (L 275)

Discussion 

Regarding the iron intake, there were no other iron supplementation programs in the research area. Further, we did a follow up every month for dietary diversity and iron-rich and bio-available food consumption such as animal product (red meat, organ meat and fish). The six month follow up indicated that there was low consumption of iron-rich food and there was no significant difference between groups both for dietary diversity score and iron-rich food consumption (L 368-372).

Reviewer #3: 

This study compares haemoglobin and ferritin response (C reactive protein adjusted) in iron rich Amaranth compared to Maize in a bread format in anaemic (<110 g/L) children 2-5 y. Results indicate a positive benefit of the Amaranth on haemoglobin, anaemia, but not ferritin. This is an important study as Amaranth has a high yield and grows well in challenging conditions. It is rich in micronutrient for a plant-based food but is high in phytates which limit micronutrient bioavailability; this study show despite that Hb is improved. However, I have some major concerns.

Thank you very much for your constructive comments we considered your comments and improved our manuscript.

Response

It is true that amaranth is rich in macro and micro nutrients and also it contains high level of phytate, which can prevent the absorption of micro and macro nutrients from the food in the gastrointestinal tract. Prior to this study, we conducted a research on the effect of home processing such as socking, germinating and roasting the grain to reduce the phytate level. The result indicated that home processing of the grain had resulted in a significant reduction of phytate level from the grain amaranth. (4). Also there is another study in Ethiopia shows that fermentation had a significant effect to decrease phytate from this specific grain (5). Accordingly, in our research, we applied home processing (socking, germinating, roasting and fermenting) to reduce the high phytate level from the amaranth grain. 

Comments Abstract

1. I do think in the abstract in the objective you need to capture what you mean by processing. Amaranth grain that has been processed to reduce the effect of phytate. It is indicated later in the abstract how it was processed but it is not clear that the reason for this processing is to reduce phytates. 

Answer

Thank you for the suggestion. Now we describe the reasons of processing in the abstract part: The amaranth was processed at home level, to decrease phytate concentration which in this case involved soaking, germinating, and fermenting. The maize arm (N=50), received 150 g bread, containing processed maize (roasted and fermented) to give a similar colour and structure with amaranth bread (L: 32-35).

2. The Betas are confusing and maybe would be better presented as an adjusted mean difference with a 95% CI. A P value should be given. I assume the B is the difference between the difference for each treatment from baseline and end line 

Answer

Thank you very much for this comment. Now we presented the adjusted estimated beta coefficient of GEE result with 95%CI and P-value

Haemoglobin concentration estimate of beta coefficient was significantly higher in the amaranth arms compared with maize arm [aβ 8.9 g/L (95%CI: 3.5-14.3)], p-value <0.01. (L: 40-41) 

3. Introduction: Clear and well written. What is the primary outcome?

Answer

Thank you for this comment. The primary outcome was hemoglobin concentration and anemia prevalence, and the secondary outcome was iron deficiency anemia and iron deficiency. Now outcome variables are clearly described in the method part (L132-133).

Methods

The sample size seems based on the continuous variable haemoglobin not on anaemia seems like it is the primary outcome in the introduction. Has the effect of clustering been taken into consideration in the sample size? The sample size seems small for this type of study. 

Answer

Thank you for this comment! Yes it is true that we used hemoglobin change of children with anemia to calculate the sample as hemoglobin change directly related with anemia prevalence. The study participants for trial were children with anemia those identified at baseline survey.

Regarding to clustering during sample size calculation, we didn’t considered clustering as there is no stratum to cluster. However, later we used clusters by geographic location to ease daily management and supervision. Furthermore, we conducted the intra class correlation during analysis (ICC) test to see the effect of cluster. The result indicated that there was no significant effect of clustering on the outcome variables (213-217).

We described in the sample size calculation as follows (L 108-116).

Prior to the trial, a census was conducted and 1689 children two-to-five year-old were identified in the study area. By using simple random sampling we selected 340 children to do a cross-sectional study in order to identify children with mild and moderate anaemia. The sample size of 340 was estimated to identify anaemic children, assuming a 33% mild and moderate anaemia prevalence in South Ethiopia, precision of 0.5, and 95% confidence intervals. The sample size of the trial was calculated considering being able to detect a mean difference of 0.7g/dl in haemoglobin level. 1 standard deviation, power of 90%, alpha level of 0.05 with two sided and an anticipated attrition rate of 12%. The total sample size of the trial was 100, with 50 in each arm. Using the survey we identified 100 children with anemia but they were sparsely distributed in the study area. We therefore clustered them in to eight clusters by their geographic locations to ease the distribution of breads and improve the follow up. The eight clusters were randomly allocated into four clusters in the maize arm and four clusters in the amaranth arm, and each arm had 50 children.

4. Line 112 should g/dl preferably g/L not mg/dl Please present haemoglobin in SI units g/L not g/dl.

Answer

We converted the SI units of haemoglobin from g/dl to g/L throughout the document. 

5. Line 119 – what was the primary outcome

Answer

Thank you for this question. The primary outcome was haemoglobin concentration and anemia prevalence, and the secondary outcome was iron deficiency anemia and iron deficiency (L132-133).

6. Maize contains phytate although less iron was the maize processed similarly to the Amaranth?

Answer

Thank you for your observation. Yes it is true that maize contain less phytate as compared with amaranth grain. However, we processed the maize too in order to make the recipe/maize bread similar in color and texture with amaranth bread. We applied only roasting and fermentation to maize bread. Probably the phytate level on maize may be decreased due to processing, but we haven’t studied it in this specific research 

Maize grain was also roasted and fermented in order to make the recipe similar in colour and structure with amaranth bread (L 147-148).

7. When was the blood collected at baseline relative to the start of the trial? Was there a screening (part of the larger survey) and then a baseline measure at the beginning of the trial.

Answer

Thank you. Yes, as you stated prior to the trial, a census was conducted and 1689 two-to-five year-old children were identified in the study area. Then we conducted the survey from February 15, to March 30, 2017 to identify children with anaemia. With the survey we identified 100 children with anaemia which included in the trial then, home based bread feeding was started for both arms on April 14, 2017 (L116-122).

8. Why did you use a categorical cut off for ferritin adjustment rather than Thurnam or BRINDA, this would have given you more sensitivity? 

Answer

Thank you very much for this suggestion. We used WHO recommendation for ferritin adjustment and not the Thurnam or BRINDA. The reason for this is to prevent over adjustment because the small sample size and we had only measure CRP and had not done AGP concentration. Now we clearly defined the adjustment in the method part (174-180).

Ferritin was adjusted for inflammation based on CRP measure. The inflammatory state of each individual was classified as “healthy” if CRP<5mg/L and “with inflammation” if CRP >5mg/L. The cut-off value for ferritin level was set at <12 µg/L for those children who had CRP<5mg/L (healthy) and <30 µg/L for those children who had CRP >5mg/L (inflammation). Iron deficiency defined as ferritin concentration <12 µg/L for healthy children and <30 µg/L for children with inflammation (1, 2). Iron deficiency anemia was defined as anemia with concurrent iron deficiency (2, 3).

9. How was the bread dispensed? Did the children come daily to a village centre to receive the bread or was it packaged and sent home on a weekly basis? In the results section it looks 

Answer

Thank you. We used home-based distribution of bread. The care givers were assigned to take the bread on a daily bases to the children’s home and feed them. We described this clearly in the methods part (L: 156-162).

The bread distributers were different for the amaranth and maize arms, and there were eight caregivers, one for each cluster. Eight boxes, one for each cluster, were packed and labeled with the respective caregiver’s name. Each caregiver feed the child at their home every day under her direct supervision. In case of refusals or absenteeism, unopened bread was returned and registered at coordinator level each day. 

10. Line 205 Appendix 3 is the first of the appendices to be called out in the text where is 1 and 2.

Answer

Thank you to pointing this. We corrected the order.

11. Setting all individuals that had missing values at end line to anaemic is really just last value carried forward. Could you justify this? Clearly with regression to the mean and screening and including only those who were anaemic many of these would be non-anaemic at end line irrespective of treatment. I suppose one could presume this would be balanced across treatment. How were missing continuous outcome variables dealt with? 

Answer

Thank you very much. Now we clarified the confusion which this created. Haemoglobin concentrations were corrected for altitudes, according to WHO standards now we reported attitude adjusted haemoglobin concentration (L 171-174).

Intention-to-treat analysis and complete case analysis were used to manage the missing value of main variables: haemoglobin, ferritin, and CRP. Intention-to-treat analysis (ITT) applied by replacing the missing values of the last follow up measure from the baseline and a complete-case analysis was restricted to available information at the last follow-up measure was used to decrease miss-interpretation (L 218-222).

12. Line 203 – 205 Is this used to show that there was no effect of clustering, yet clustering is included in the GEE. In the abstract it looks like the difference of difference was used to look at treatment effects. However, it looks like baseline values were included in the model. I get a bit lost in the stats there is a baseline measure and one follow-up measure? 

Answer

Thank you for pointing this comment. Now we omitted clustering from multivariate analysis since there was no change on the significance result. Regarding the mean difference, now we report only estimated beta coefficient to avoid confusion to compare the result haemoglobin difference between groups (L 292-297).

13. Perhaps more clearly delineate how categorical and continuous dependent variable were dealt with. 

Answer

Thank you. We have described how we handled the categorical and continuous variable in the methods. 

Generalized estimating equations (GEE) procedure was used for analysis of repeated measurements (hemoglobin ferritin and CRP) once at baseline and once at last follow up. Hemoglobin was not normally distributed and included as continuous variable. Adjusted ferritin with CRP was used to categorize iron deficiency anemia and iron deficiency so that it was included as categorical variable (L: 224-227).

14. Table 2 and 3. It is not appropriate to include P-Values when comparing treatment groups because if and difference exist they must be a chance finding. I believe you have adjusted for these in the models anyway which is appropriate. Is all the information required in Table 2 and 3 necessary? It would be appropriate to include the severity of anaemia by treatment group and low ferritin by treatment group.

Answer

Thank you for the comments. We removed less relevant information from the table and we included the magnitude of baseline anemia and iron deficiency prevalence between treatment groups (Line 259-264)

Please see Table 5 Magnitude of anemia and iron deficiency anemia distribution between groups (Line 259-264)

15. The participants needed to anaemic at baseline (<110 g/L) for inclusion yet the mean Hb for maize is 111 g/L with the upper bound of the CI at 114 g/L. Also, the number of children is highly clustered around the cut-off 108-114 g/L. Is anaemia prevalence the correct outcome. This all leads me to believe that there was a screening and a baseline blood taken but this is not clear. Would Hb be more appropriate with a lot of people not anaemic at baseline how was dealt with in the analysis

Answer

Thank you. Yes your concern is right as the variation is due to altitude adjustment to hemoglobin level which we missed in our report. The altitude of the study area is which requires adjustment for hemoglobin now we presented the altitude adjusted Hb (L167-170).

Anaemia was classified based on altitude-adjusted haemoglobin as normal (≥110.0 g/L), mild (100.0–109.0 g/L), moderate (70.0 –99.0 g/L), or severe (<70.0 g/L). Children with severe anaemia (<70.0 g/L) identified in the survey were excluded from the trial, as they were referred for medical treatment (N=1). Haemoglobin concentrations were corrected for altitudes, according to WHO standards to diagnosis anaemia thus, we subtracted 8.0g/L from the original reading

16. I am really confused on the ferritin. You need to adjust the ferritin values using Thurnam or Brinda and then use some sort of linear regression. Table 6 doesn’t make sense to have three cut-offs, it should be the percentage with low iron based on the appropriate cut-off for ferritin. 

Answer

Thank you for pointing this. We used WHO recommendation for ferritin adjustment. Since our sample size is small and had only measure of CRP and as we did not analyse AGP concentration, we didn’t used internal correction factor to prevent over adjustment.

Ferritin was adjusted for inflammation based on CRP measure. The inflammatory state of each individual was classified as “healthy” if CRP<5mg/L and “with inflammation” if CRP >5mg/L. The cut-off value for ferritin level was set at <12 µg/L for those children who had CRP<5mg/L (healthy) and <30 µg/L for those children who had CRP >5mg/L (inflammation). Iron deficiency defined as ferritin concentration <12 µg/L for healthy children and <30 µg/L for children with inflammation (1, 2). Iron deficiency anemia was defined as anemia with concurrent iron deficiency (2, 3) (L: 174-180).

The result presented on Table 10 and Table 11 please see (L 301-314) 

Discussion

17. I don’t think Amaranth would contain any B12 Line 343. Also did you consider the presence of sickle cell or other genetic Hb disorders 

Answer

Thank you. We accept the comment and we put it as a limitation of the study which did not measure different causes of anaemia. 

This study does not assessed other nutrient-related anaemia, such as folic acid deficiency, vitamin A deficiency, and copper deficiency as well as does not considered genetic related haemoglobin disorder even though it is not common in our setup (L:381-384).

1. Rosalind GS. Principles of Nutrtional Assessment Second ed. New York: Oxford University Press,Inc; 2005.

2. World Health Organization. Serum ferritin concentrations for the assessment of iron status and iron deficiency in populations vitamin and mineral nutrition information system Geneva: WHO/NMH/NHD/MNM; 2011. Available at: https://www.who.int/vmnis/indicators/serum_ferritin.pd

3. World Health Organization. C-reactive protein concentrations as a marker of inflammation or infection for interpreting biomarkers of micronutrient status vitamin and mineral nutrition information system. Geneva WHO/NMH/NHD/EPG; 2014. Available at: https://www.who.int/nutrition/publications/micronutrients/indicators_c-reactive_protein/en/

4. Zebdewos A, Singh P, Birhanu G, Whiting SJ, Henry CJ, A Kebebu. Formulation of complementary food using amaranth, chickpea and maize improves iron, calcium and zinc content. ajfand. 2015;15(1)(1684 5374). available from: https://www.ajol.info/index.php/ajfand/article/view/124799

5. Amare E, Mouquet-Rivier C, Rochette I, Adish A, Haki GD. Effect of popping and fermentation on proximate composition, minerals and absorption inhibitors, and mineral bioavailability of Amaranthus caudatus grain cultivated in Ethiopia. J Food Sci Technol. 2016;53 (7):2987-94. http://doi.org./10.1007/s13197-016-2266-0 PMID: 27765968

---

## [Decision Letter · Decision Letter 1]

2 Sep 2020

Efficacy of processed amaranth-containing bread compared to maize bread on hemoglobin, anemia and Iron deficiency anemia prevalence among two-to-five year-old anemic children in Southern Ethiopia: A cluster randomized controlled trial

PONE-D-19-33677R1

Dear Dr. Orsango,

We’re pleased to inform you that your manuscript has been judged scientifically suitable for publication and will be formally accepted for publication once it meets all outstanding technical requirements.

Kind regards,

Timothy J Green

Guest Editor

PLOS ONE

Additional Editor Comments (optional):

Thank-You You have addressed the reviewers comments well.

Reviewers' comments:

Reviewer's Responses to Questions

**Comments to the Author**

1. If the authors have adequately addressed your comments raised in a previous round of review and you feel that this manuscript is now acceptable for publication, you may indicate that here to bypass the “Comments to the Author” section, enter your conflict of interest statement in the “Confidential to Editor” section, and submit your "Accept" recommendation.

Reviewer #2: All comments have been addressed

2. Is the manuscript technically sound, and do the data support the conclusions?

Reviewer #2: Yes

3. Has the statistical analysis been performed appropriately and rigorously? 

Reviewer #2: Yes

4. Have the authors made all data underlying the findings in their manuscript fully available?

Reviewer #2: Yes

5. Is the manuscript presented in an intelligible fashion and written in standard English?

Reviewer #2: Yes

6. Review Comments to the Author

Reviewer #2: Thank you for addressing all comments, this is a very good addition to the literature in this area of study.

7. PLOS authors have the option to publish the peer review history of their article (what does this mean?). If published, this will include your full peer review and any attached files.

Reviewer #2: No

---

## [Editor Report · Acceptance letter]

15 Sep 2020

PONE-D-19-33677R1 

Efficacy of processed amaranth-containing bread compared to maize bread on hemoglobin, anemia and Iron deficiency anemia prevalence among two-to-five year-old anemic children in Southern Ethiopia: A cluster randomized controlled trial 

Dear Dr. Orsango:

I'm pleased to inform you that your manuscript has been deemed suitable for publication in PLOS ONE. Congratulations! Your manuscript is now with our production department. 

Kind regards, 

on behalf of

Dr. Timothy J Green 

Guest Editor

PLOS ONE